# Enhanced Anti-Allergic Activity of Milk Casein Phosphopeptide by Additional Phosphorylation in Ovalbumin-Sensitized Mice

**DOI:** 10.3390/molecules24040738

**Published:** 2019-02-19

**Authors:** Ntshepisa Lebetwa, Yuta Suzuki, Sachi Tanaka, Soichiro Nakamura, Shigeru Katayama

**Affiliations:** 1Interdisciplinary Graduate School of Science and Technology, Shinshu University, 8304 Minamiminowa, Ina, Nagano 399-4598, Japan; nlebetwa@gmail.com (N.L.); snakamu@shinshu-u.ac.jp (S.N.); 2Department of Agricultural Research, Animal Production and Range research Division, Ministry of Agriculture, Private bag 0033, Gaborone BO320, Botswana; 3Faculty of Agriculture, Shinshu University, 8304 Minamiminowa, Ina, Nagano 399-4598, Japan; 18as207a@shinshu-u.ac.jp (Y.S.); tanakasa@shinshu-u.ac.jp (S.T.)

**Keywords:** phosphopeptide, phosphorylation, anti-allergic activity, immunomodulation

## Abstract

The proteolytic digest of milk casein, known as casein phosphopeptide (CPP-III), exhibits diverse biological activities, including calcium absorption and antioxidant activities. We hypothesized that the additional phosphorylation of this peptide can enhance its immunomodulatory activity such as suppression of allergy-associated cytokine and antigen-specific immune response. This study was conducted to assess whether oral intake of additionally phosphorylated CPP-III (P-CPP) attenuates ovalbumin (OVA)-induced IgE-mediated allergic reactions because of the additional phosphate groups. Female BALB/c mice were intraperitoneally sensitized with OVA twice at intervals of 14 days and then orally fed native CPP-III (N-CPP), P-CPP, and dephosphorylated CPP-III (D-CPP) for 6 weeks. Next, the mice were orally challenged with 50 mg of OVA. Oral administration of P-CPP suppressed total and specific IgE levels in the serum. Mice fed P-CPP exhibited low levels of OVA-specific IgG1 and increased OVA-specific IgG2a. P-CPP also suppressed IL-4 production, while D-CPP showed similar a level compared to that of the control. Further, P-CPP increased the population of the T follicular helper (Tfh) cell in the spleen. These results suggest that additional phosphorylation of CPP can enhance the attenuation of allergen-specific IgE-modulated allergic reactions in a murine food allergy model.

## 1. Introduction

Food is essential for nutrition and health, but depending on dietary habits it is considered to be a major determinant of most chronic diseases including food allergies. Severe IgE food-induced allergic reactions are responsible for a variety of symptoms involving the skin, gastrointestinal tract, and respiratory tract [1]. Allergic diseases are caused by the failure to develop and sustain oral immune tolerance to well-known and harmless allergens including aeroallergens and foods [2]. An effective tolerogenic immune response towards allergens is a key factor in preventing the pathogenesis of allergic diseases. Food allergy affects between 5% and 7.5% of children and between 1% and 2% of adults [3]. This dramatically increases the burden of allergic diseases and results in substantial financial costs incurred by affected individuals. These allergies also impact the world economy by increasing economic costs related to health care and the loss of productivity at workplaces. Although therapies for allergic diseases have improved over the past several years to reduce inflammatory processes and provide symptomatic relief, these therapies are non-curative. Therefore, it is an important research subject to search and develop food dietary compounds that can safely and effectively suppress or inhibit the formation of antigen-specific IgE; prevent severe IgE-mediated, food-induced anaphylactic reactions; or promote oral immune tolerance [4].

Bovine milk protein is a major source of valuable bioactive peptides and is released upon enzymatic hydrolysis during gastrointestinal transit or upstream during food processing [5]. The enzymatic or gastrointestinal digestion of milk protein results in the production of functionally and physiologically active peptides. Meisel et al. found that the bioactivities of peptides in milk proteins remain latent until the peptides are released and activated [6,7]. Similarly, Park and Nam suggested that most of the bioactivities of milk proteins are absent or incomplete in the original native protein, but full activities are observed upon proteolytic digestion, which releases and activates the bioactive peptides from the original protein [8].

Casein phosphopeptides (CPPs) are phosphorylated peptides produced by the proteolysis of calcium-sensitive caseins and possess a phosphoserine-rich region [9]. These peptides have multiple bioactive functionalities. Various researchers have widely reported the bioactivities of CPP, particularly the commercial CPP-III, which consists mainly of β-casein (1–28) and α-s2 casein (1–32). Previously, other researchers reported that CPP-III enhances intestinal IgA by promoting the production of interleukin (IL)-6 in mice [10,11] and CPP-III exerts immune-enhancing activities by stimulating the proliferation of mouse spleen cells [12]. However, they did not fully explore its mechanism of action in relation to its phosphate groups.

In this study, we investigated the anti-allergic activities of CPP-III in ovalbumin (OVA)-sensitized BALB/c mice as a mouse model of egg allergy. Chicken egg OVA is recorded as the most common cause of food-induced allergic reaction. Sensitization with OVA can trigger some robust and life-threatening immune hypersensitivity with symptoms comprising of anaphylaxis, atopic dermatitis and inflammation of the esophagus [13]. The effects of additional phosphorylation and dephosphorylation on CPP-III activity were also assessed to clarify the role of the phosphate groups.

## 2. Results

### 2.1. Phosphorylation and Dephosphorylation of CPP

Phosphorylated CPP-III (P-CPP) was prepared by dry-heating in the presence of pyrophosphate buffer. After dry-heating at 85 °C for 5 days, the phosphorous content of P-CPP was 18.8 μg/mg of protein, which was significantly higher than that of native CPP-III (N-CPP) with 9.8 μg/mg of protein. In contrast, dephosphorylated CPP-III (D-CPP) was prepared by incubating CPP-III with bovine alkaline phosphatase at 37 °C for 24 h to remove the phosphate groups. The phosphorous content of D-CPP was below the detection limit.

### 2.2. Effects of Oral Administration of P-CPP and D-CPP against OVA-Induced IgE and Allergic Reactions

All mice except naïve mice were sensitized twice by intraperitoneal immunization with 50 μg of OVA at an interval of 2 weeks prior to the oral challenge with 50 mg of food allergy model OVA and only PBS in the naïve group. Mice were then provided ad libitum oral access to feed containing 0.05% (*w*/*w*) N-CPP, P-CPP, and D-CPP and commercial feed for the naïve group and sham-treated control group (Figure 1). Temperature changes were recorded immediately after the oral challenge with OVA and significant body temperature decreases of −1.47 ± 0.94 °C and −1.67 ± 1.71 °C were observed at 10 and 20 min in the sham-treated control and D-CPP groups, respectively; the naïve, N-CPP, and P–CPP groups showed body temperature changes of 0.93 ± 1.04, −0.37 ± 0.47, and −0.43 ± 0.43 °C, respectively (Figure 2A). Systemic anaphylactic symptoms were recorded in the sham-treated control and D-CPP groups after 40 min, which showed average scores of 1.67 ± 0.56 and 2.0 ± 0.37, respectively. In contrast, the average scores were significantly lowered in the N-CPP and P-CPP groups to 0.33 ± 0.21 and 0.66 ± 0.21, respectively, compared to the sham and D-CPP group (*p* < 0.05) (Figure 2B). No symptoms were observed in the naïve group. These results suggest that oral treatment with P-CPP was able to reduce the allergic reactions induced by OVA exposure as witnessed by reduced allergic score and stable temperature.

We next investigated whether the development of the antigen-specific immune response could be regulated by oral treatment with P-CPP. Mice orally treated with P-CPP showed significantly suppressed production of total IgE antibody in the blood serum as compared to the sham-treated group (*p* < 0.05), and the D-CPP treated mice group exhibited less suppression towards total IgE compared to P-CPP (*p* < 0.01) (Figure 3A). A significant decrease in production was also observed in the levels of OVA-specific IgE in the serum of mice treated with P-CPP than the sham-treated mice *(p* < 0.01) (Figure 3B). Mice in the P-CPP treated group exhibited a slight increase in levels of total IgA compared to the sham-treated, N-CPP, and D-CPP treated groups. The OVA-specific IgA levels were significantly increased in P-CPP fed mice compared to the sham-treated group (*p* < 0.01), while there was a less significant increase in D-CPP fed mice compared to the P-CPP fed mice (Figure 3C,D). The secretion of OVA-specific IgG1 levels in the serum of the P-CPP treated group were significantly lower than in the control group (*p* < 0.05), however, OVA-specific IgG2a was remarkably increased in the P-CPP group compared to in the sham-treated control group (*p* < 0.05). The D-CPP treated group showed significantly less secretion of IgG2a than the P-CPP group (*p* < 0.05) (Figure 3E,F). The results suggest that the phosphorylation of CPP-III might attenuate the allergic reactions against OVA allergen more than the non-phosphorylated CPP-III. 

### 2.3. Effects of Orally Fed Phosphorylated and Dephosphorylated CPP-III on OVA-Induced Cytokine Production In Vitro

We next measured the production of IFN-γ (Th1-associated cytokine) and IL-4 (Th2-associated cytokine) in spleen (SP) cells to determine whether oral treatment with N-CPP shifted the Th1/Th2 balance to a Th1 dominant response. The IFN-γ secretion level in culture supernatants was significantly increased by oral treatment with P-CPP, more than the N-CPP and D-CPP groups (*p* < 0.05). (Figure 4A). In contrast, oral administration of P-CPP to the mice significantly inhibited IL-4 cytokine production more than the N-CPP and D-CPP treated groups (*p* < 0.01) (Figure 4B). Stimulation of cultured cells by phosphorylated OVA (P-OVA) further reduced IL-4 compared to in the native OVA-stimulated control group (data not shown). To further elucidate and characterize the differentiation of naïve T cells to either Th1 or Th2 cells after oral treatment with P-CPP, we tested for the gene expression levels of the transcription factor GATA-3 and IL-4 cytokine in isolated SP cells. There was a significant reduction in expression of both GATA-3 and IL-4 mRNA in P-CPP fed mice as compared to N-CPP and D-CPP (*p* < 0.05). IL-4 is reported to induce B-cell class switching to IgE and also decreases Th1 cells production, while GATA-3 is selectively expressed in Th2 but not in Th1 and is also important in the expression of IL-4 in T cells by transactivation of the IL-4 promoter. The results suggest their suppression promoted the shift from Th2 immune response to Th1 immune response. In addition, the higher serum OVA-specific IgE level in sham-treated control group might be attributable to the higher IL-4 level. 

### 2.4. Effects of Orally Fed Phosphorylated CPP-III on Differentiation and Population Changes of Regulatory T (Treg) Cells and T Follicular Helper (Tfh) Cells

We evaluated the effect of orally administered P-CPP on the differentiation level of Treg and Tfh in the supernatants of SP and payer’s patch (PP) cells in the murine allergy model. Flow cytometric analysis was conducted after 6 weeks of P-CPP feeding. We counted CD4+ CD25+ Foxp3+ cells as Treg. The results showed that the population of Treg in the cultured SP cells of the P-CPP treated group was higher (16.6 ± 0.8%) than that in the sham-treated control group (12.9 ± 0.3%) (Figure 5A). Treg was comparatively lower in PP cultured cells compared to in cultured SP cells. The percentage of cells in the P-CPP treated group (9.7 ± 0.6%) was higher than that in the sham-treated control group (7.1 ± 0.5%) but lower than that in the N-CPP and D-CPP groups (10.3 ± 0.2% and 10.2 ± 0.4%, respectively) (Figure 5B). We also determined the proportion of Tfh cell population changes in the SP and PP cells after oral administration of P-CPP. The population of Tfh counted as CD4+ CXCR5+ CD279 cells in SP cells of mice orally fed P-CPP was higher (3.0%) than that in the sham-treated control group (1.8%) (Figure 6A). The same trend was observed in PP cells (3.6%) of the Tfh population for P-CPP (2.3%) in OVA-treated control mice (Figure 6B). These results suggest that the amplified Tfh populations might be caused by phosphate groups of CPP, which suppressed the onset of allergic diseases by downregulating Th2 immune response.

## 3. Discussion

In our present study, we established that the oral intake of highly phosphorylated casein phosphopeptide could attenuate type-I allergic response in an OVA model mice through the induction of Tfh. Bioactivities of casein hydrolysates have been widely studied. In a previous report, Bamdad et al. showed that casein hydrolysates exhibited anti-inflammatory and antioxidant properties by significantly reducing nitric oxide and suppressing the synthesis of pro-inflammatory cytokines (TNF-α and IL-1β) in lipopolysaccharide stimulated RAW 264.7 macrophage cells [14].

In this study, after the oral challenge with OVA, a marked decrease in body temperature or hypothermia was observed in the Sham and D-CPP treated group, while it was practically unchanged in the naïve N-CPP and P-CPP groups. Mice provided P-CPP also exhibited a lower allergic score as compared to the sham-treated and D-CPP groups. This similar IgE systemic anaphylaxis reaction was reported by Makabe-Kobayashi et al. who linked it to the activities of mast cell-derived histamine using Histidine decarboxylase gene knockout (HDC−/−) mice, which lacks histamine [15]. 

IgE antibodies are the mediators of most food allergic reactions and overall, food-based allergies are characterized as IgE-mediated hypersensitivity reactions [16]. Burton and Oettgen emphasized that IgE binds to two main receptors, the high affinity FcεRI and FcεRII. This interaction of IgE with its receptors expressed on mast cells is considered to play a significant role in maintaining a sensitized state in food allergic patients by focusing on stimulating memory T and B cell responses; it is also considered to amplify the Th2 and IgE response [2,17]. As such, discoveries of safer and effective methods to inhibit allergen specific-IgE production are vital in controlling type-I allergic reactions. After oral feeding of P-CPP for 6 weeks, we noticed a significant decrease in levels of total IgE and OVA-specific IgE in serum; in contrast there was an increase in total IgA and OVA-specific IgA. IgA is reported to mediate both pro- and anti-inflammatory effects in innate immune cells [18]. The protective effect of IgA antibodies against infectious antigens in the gut is well documented but little is elucidated on the role of IgA in food allergy. Previously, some researchers showed that the oral ingestion of different preparations of commercial casein phosphopeptides, including CPP-I and CPP-III, by mice enhanced intestinal and milk IgA and IL-6 expression [9,10,11]. Recently, Kiewiet et al. emphasized that the administration of different protein hydrolysate like common carp egg hydrolysate in vivo can increase the secretion level of IgA and IgA+ cells. They point out that IgA functions in clearance of toxins when it is released in the gut and is easily measured in feces [19]. This suggests that the peptide might not enhance only IgA, but also various immunoglobulins in the immune system including the T cell family. In our previous study, we found that P-CPP enhanced antiviral activities against feline calicivirus infection by inducing antiviral cytokines such as IFN-α and IFN-β. Phosphorylated CPP-III exhibited stronger antiviral activities than its native and dephosphorylated forms. We next demonstrated that the stronger antiviral activities of P-CPP were dependent on the negative electrostatic charge of the phosphate groups attached to the amino acid chain [20]. An overview of recent researches [19] indicated that dietary ingredients, especially immunomodulatory protein hydrolysates, exhibit the capacity to attenuate food allergic reactions. Included in this protein hydrolysate are milk casein hydrolysates, egg yolk digests, yellow pea, shark protein, and soy bean hydrolysates. They exhibit their immunomodulatory properties through the upregulation of pro-inflammatory cytokines (IL-10, TNF-α, IL-6 and IFN-γ), increase of IgA+ cells, elevation of secretory IgA in the gut, increase in Treg in the spleen, and reduction of IgE production [21,22,23]. We also observed that the secretion of Th2-driven IgG1 was markedly inhibited by oral administration of P-CPP, whereas the secretion of Th1-driven IgG2a was significantly increased. These hypo-allergic properties of some milk protein hydrolysates including caseins and whey is in agreement with studies by Kiewiet et al. and Pan et al. who reported on the immunomodulatory and hypoallergenic properties of milk protein hydrolysates through enhancing regulatory T and B cell frequencies [22,24].

We then determined the cytokine levels in culture supernatants of P-CPP administered mice splenocytes. We observed that the oral ingestion of P-CPP by mice profoundly increased the splenocytes production of Th1-related IFN-γ and significantly suppressed the secretion of Th2-related IL-4 cytokine. Naturally, IgE synthesis is considered to be due to the development and activation of Th2 cells and B cells. Native CD4^+^ T cells are induced to differentiate into helper Th2 cells that sequentially promote the production of IgE via the predominant production of cytokines such as IL-4. IL-4 plays an important role in inducing class switching of the IgE isotype and its production [25]. Excessive secretion of IL-4 by Th2 is associated with increased IgE levels and subsequent allergic reactions. In contrast, based on the production patterns of cytokines by helper T cells, a Th1 type immune response occurred that mainly involved the secretion of cytokines such as IFN-γ, which inhibit IgE and IgG1 secretion and enhance IgG2a secretion [26]. Th1 cytokines are possibly the ones that inhibited the secretion of IgG1 and enhanced that of IgG2a above. Helper T cells are generally induced by different cytokines and controlled by transcription factors. They are reported to be classified into Th1 and Th2, which is determined by the signature of cytokines produced [27]; these T cells are known to maintain a well-balanced relation in the modulation of cytokine secretion to keep homeostasis in the host [4]. In order to characterize Th2 response, we measured the effect of P-CPP ingestion on gene expression levels of the transcription factor GATA-3 and IL-4 cytokine. Both GATA-3 and IL-4 mRNA levels in total spleen cells were significantly inhibited by oral treatment with P-CPP. GATA-3 is a key transcription factor that is selectively expressed at high levels in Th2 cells [28]. It is conventionally regarded as a transcription factor that drives and regulates the differentiation of Th2 from naïve CD4^+^ lymphocytes [27,29], and it is reported to significantly inhibit the expression of Th1-related cytokine IFN-γ while upregulating the gene expression of Th2-related IL-4 and IL-5 [30]. Th2 related cytokines are the mediators of allergic inflammation. Previously, Finotto et al. found out that the blockade of GATA-3 expression in the lung by oligonucleotide leads to the suppression of airways inflammation [31]. Our results suggest that administration of P-CPP skewed the balance from Th2 towards Th1 dominance. The development and activation of Th2 cells and B cells is thought to increase IgE synthesis and IL-4 and IL-5 cytokine production [4]. This shift also confirms that P-CPP has a modulating ability on Th1/Th2 balance to down-regulate Th2 response. 

We also observed the induction and differentiation of precursor cells into Tfh cells, leading to an increase in their numbers in the SP and PP cells. These cells are specialized providers of T cell help to B cells in order to support their activation, expansion, differentiation, and formation of germinal centers (GC). In our study, feeding of P-CPP to mice significantly increased the population of Tfh both in supernatants of cultured SP and PP cells. Tfh cells were reported to promote the secretion and production of IgA by enhancing the differentiation of IgA+ B cells [32]. This increase in the Tfh cell population may have been responsible for the suppression of Th2-induced allergic reactions, including IgE production, following sensitization with OVA. Previous studies have also demonstrated that Tfh secreted IL-21 can suppress T-cell production of Th2 cytokines, mast cell degranulation and inhibit the production of allergen-specific IgE [33,34]. The inhibition of class switching to IgE by IL-21 was proven by the production of high levels of IgE by IL-21R-deficient mice as compared to those with IL-21 receptor [35]. Achieving a balance and modulation between the Th1/Th2 immune response is regarded as the best immunotherapy strategy for allergic diseases [36]. While Tfh cells have been notably known to produce IL-21, which inhibits class switching to IgE, there are other existing reports of Tfh cells producing other cytokines, among them IL-4, which may determine the antibody produced [36,37]. Recent studies which are consistent with these findings have observed the development of IL-4-producing Tfh cells and investigated how they might be playing a critical role in IgE production in peanut allergy [38,39]; in addition, they also discovered the type 2 subset (Tfh2) within the Tfh which are considered as the major player that secretes IL-4 and promotes isotype switching to IgE after allergen exposure and during intestinal helminth infection [40]. Despite all these recent findings, the functions of Tfh remain less clear and even though it is indicated from most researches that their secretion of IL-4 is necessary for IgE production, it does not definitely rule out the contribution of Th2 cell-derived IL-4 in this particular immune response. As such, further investigation is required to understand the development and/ or functions of Tfh and its relationship with effector Th1 and Th2. 

Peptides liberated from milk by enzymes were demonstrated to possess immunomodulatory properties [41] and should be considered as potential modulators of various regulatory processes in the body. Casein phosphopeptides are released from casein through proteolytic digestion in the small intestine. They were reported to resist further digestion by intestinal proteinases or bacterial proteinases in the digestive tract and accumulate in the most distal part of the small intestine [42,43]. A review by Scherer et al. pointed out that allergen-specific immunotherapy also includes immunotherapy with modified proteins that are designed to be hypoallergenic to reduce the risk of immune reactions towards food proteins [44]. Ueno et al. developed an edible hypoallergenic casein hydrolysate which reduced the antigenicity for casein-specific antibodies while retaining the immunogenic epitopes. They achieved this through digesting casein at alkaline pH [45]. Recently, Kim et al. using intact casein as an allergen demonstrated that mesenteric lymphnode (MLN) IL-10 producing CD5^+^ B cell suppressed casein-induced allergic responses in mice [46]. It was deduced from these findings by Mensiena et al. that casein might have been digested in the intestine of mice, after which the newly formed peptides derived from casein increased the Bregs which induced oral tolerance [19]. These findings suggest the high possibility of casein and its hydrolysates including P-CPP’s hypoallergenic properties against cow’s milk allergy. Even though proteins and peptides exhibit these bioactive potentials, there is a possibility of exerting some allergic reactions. Information about their allergenicity is crucial in their all-inclusive evaluation. Reddi et al. demonstrated that CPP suppressed the release of histamine and tryptase from mouse mast cells, suggesting the attenuation of mast cell degranulation by CPP [47]. On the other hand, Bernard et al. when evaluating the influence of phosphorylation on the immunoreactivity of caseins found out that major phosphorylation sites in caseins are an important allergenic epitope [48]. In this study, we observed no allergic symptoms in P-CPP fed mice, but further investigation will need to clarify the in vivo sensitizing capacity of P-CPP.

In the past, several studies have assessed the modification of allergenic proteins to be used in immunotherapeutic treatments. Of these modifications, Maillard-type glycosylation demonstrated some efficacy in modifying the surfaces of target proteins [49]. In this method, the attachment of polysaccharides such as glucomanan and xyloglucan through maillard reaction exhibited some masking of IgE epitopes of allergenic proteins resulting in reduced IgE binding capacity in the sera of allergic patients. This conjugate has also been shown to shift the Th1/Th2 balance in spleen towards a Th1-dominated immune response [50]. Polysaccharide moieties by nature have long side chains, as such, this might hamper their ability to reach and mask epitopes located at the interior of proteins. The addition of phosphate groups to a protein might not only function as a more effective method to mask the epitope site of allergenic proteins but also as a strong enhancer of immunomodulation as evidenced by its previous promotion of type-I IFN secretion [20]. 

A recent study showed that phosphorylated buckwheat hypoallergenic protein P-Fag e 2 more strongly suppressed some Th2-induced allergic responses in a murine model of buckwheat allergy compared to its native form [36]. This suppressive capability occurred because of the increased secretion of total and specific IgA and the induction of Tfh cells regulated by dendritic cell-derived IL-6. Thus, the increase in IgA and induction of Tfh and Treg suppressed OVA-specific and total IgE by P-CPP possibly through a similar mechanism as used by P-Fag e 2. In this study, we have not evaluated the immunomodulatory activities of P-CPP against cow’s milk allergy and its hypoallergenecity. Further studies will be needed to elucidate that and the specific pathways involved in shifting the immune response.

## 4. Materials and Methods

### 4.1. Materials

CPP-III was kindly provided by Meiji Food Materia Co., Ltd. (Tokyo, Japan). Antibodies and reagents used for flow cytometry analyses were purchased from BD Biosciences Co., Ltd. (San Jose, CA, USA). All other reagents were of analytical grade.

### 4.2. Preparation of Phosphorylated and Dephosphorylated CPP-III

CPP-III was phosphorylated as previously described in our paper [20]. Briefly, CPP-III was dissolved at 1 mg/mL in 0.1 M sodium pyrophosphate buffer (pH 4.0) and then lyophilized. The lyophilized sample was dry-heated at 85 °C for 5 days. The dry-heated sample was dissolved in deionized water and dialyzed to remove free phosphate for 2 days using regenerated cellulose membranes (MWCO 1000, Spectrum Laboratories, Rancho Dominquez, CA, USA). This sample was then lyophilized to obtain phosphorylated CPP-III. In contrast, CPP-III was dephosphorylated by incubation for 24 h with bovine alkaline phosphatase at 37 °C as reported previously [51]. 

### 4.3. Mice

Female BALB/c mice were purchased from Charles River (Tokyo, Japan) at 5 weeks of age. All mice were housed in groups of six per cage and the animal room was maintained at a controlled temperature of (20–24 °C), humidity (40–70%), with alternating 12-h/12-h light–dark cycles (lights on at 8:00 am and off at 20:00 pm. All animal experiments were performed in accordance with the animal experiment protocol that was approved by the Institutional Animal Care and Use Committee of Shinshu University.

### 4.4. OVA-Sensitized Allergic Model Mice

Model mice sensitized with OVA were prepared as previously described [52,53]. The mice were divided into five groups (six mice per group). For the OVA sham-treated control, N-CPP, P-CPP, and D-CPP groups, naïve mice were immunized with 50 μg OVA dissolved in 100 μL of phosphate-buffered saline (PBS) and alum adjuvant by intraperitoneal injection. Mice were then boosted by intraperitoneal injection with 50 μg OVA and alum adjuvant at 14 days after the initial immunization. On day 21, after confirming the elevated serum levels of specific IgE antibodies via enzyme-linked immunosorbent assay (ELISA), the mice were provided ad libitum access to homemade feed containing 50 mg N-CPP, P-CPP, or D-CPP per 100 g MF pellet for 6 weeks. Mice were then orally challenged with 50 mg of OVA on day 72. The body temperature of the mice was measured from the rectum at 0, 10, 20, 30, 40, 50, and 60 min using a Weighing environment logger (AD1687, A&D Company, Limited, Tokyo, Japan). Mice in the naïve control group were provided with a commercial pellet diet on the same schedule. All mice were sacrificed by asphyxiation with CO_2_, and their sera and small intestines were harvested. For cytokine analyses, SP cells harvested from mice in the naïve, OVA sham-treated control, N-CPP, P-CPP, and D-CPP groups were incubated with 50 μg/mL (final concentration) OVA at 37 °C in a humidified atmosphere with 5% CO_2_ for 72 h. Cells in the naïve control group were incubated with PBS rather than with allergens.

### 4.5. Allergic Score

Mouse allergic symptoms were observed and scored from 40 to 90 min post-oral challenge dose as previously described by Li et al. [54]. Anaphylactic symptoms were scored as follows: 0, no symptoms; 1, scratching and rubbing around the nose and head; 2, puffiness around the eyes and mouth, diarrhea, pilar erecti, reduced activity, and or decreased activity with an increased respiratory rate; 3, wheezing, labored respiration, and cyanosis around the mouth and tail; 4, no activity after prodding or tremor and convulsion; and 5, death.

### 4.6. ELISA Analysis for OVA-Specific IgE, IgA, Total IgE, IgA, and OVA-Specific IgG1 and IgG2a

After collection, blood samples were incubated at room temperature for 1 h. The samples were then centrifuged at 1000× *g*, 4 °C for 15 min to collect the serum. The levels of total IgE and IgA, OVA-specific IgE and IgA, IgG1, and IgG2a in the blood serum were measured by sandwich ELISA. Horseradish peroxidase (HRP)-labeled antibodies and hydrogen peroxide with *o*-phenylenediamine were used as substrates. Anti-mouse IgE and IgA and HRP-conjugated anti-mouse IgE and IgA antibodies used in this experiment were purchased from Pierce (Rockford, IL, USA). OVA-specific IgE, IgA, IgG1, and IgG2a were detected as described for total IgE and IgA, except for OVA-specific IgE and IgA, the coating antigen was substituted with OVA (1 mg/mL).

### 4.7. ELISA Quantification of Cytokine Levels in Splenocytes

The levels of cytokines secreted into the murine SP cell culture supernatants were measured by sandwich ELISA using the following HRP-conjugated antibodies and hydrogen peroxide with *O*-phenylenediamine as the substrate: rat anti-mouse IL-4 and IFN-γ antibodies, biotinylated rat anti-mouse IL-4 and IFN-γ antibodies, and streptavidin-HRP conjugate (all purchased from Peprotech, Rocky Hill, NJ, USA).

### 4.8. GATA-3 and IL-4 Gene EXPRESSION Analysis by Quantitative PCR (qPCR)

The total RNA was isolated using Invitrogen, TRizol reagent (Thermo Fisher Scientific, Inc., Waltham, MA, USA) in accordance with the manufacturer’s instructions. cDNA was synthesized from total RNA samples using ReverTra Ace (Toyobo, Osaka, Japan). Gene expression levels were quantified with a Kapa SYBR Fast qPCR kit (Kapa Biosystems, Woburn, MA, USA) and qPCR was performed with a Thermal Cycler Dice Real time system (Takara, Shiga, Japan). Fold changes in the relative mRNA expression level for each gene were normalized to that of housekeeping gene Glyceraldehyde-3-phosphate dehydrogenase (GAPDH). The expression in experimental groups was expressed as a fold of control using the comparative C_t_ method. Forward and reverse primer sequences were: IL-4: 5′CGAAGAACACCACAGAGAGTGAGCT-3′ (forward) and 5′GACTCATTCATGGTGCAGCTTATCG-3′ (reverse); GATA-3: 5′CTACCGGGTTCGGATGTAAGTC-3′ (forward) and 5′GTTCACACACTCCCTGCCTTCT-3′ (reverse); and GAPDH: 5′ACAACTTTGGCATTGTGGAA-3′ (forward) and 5′GATGCAGGGATGATGTTCTG-3′ (reverse).

### 4.9. Flow Cytometric Analysis of Treg and Tfh Cell Populations

We conducted flow cytometry analysis to assess the changes in the cell populations including the number of Treg and Tfh cells. After oral administration of the samples to OVA-sensitized mice for 6 weeks, SP and PP cells were collected. The cells were stained with fluorescein isothiocyanate (FITC)-labeled anti-CD4 (1:100, BD Biosciences) and phycoerythrin (PE)-labeled anti-CD25 (1:100, BD Biosciences). After surface staining, the cells were washed, fixed, permeabilized, and stained for intracellular Foxp3 using Alexa-Fluor-647-labeled anti-Foxp3 (1:100, BD Biosciences) monoclonal antibodies and a mouse Foxp3 buffer set (BD Biosciences) according to the manufacturer’s instructions for Treg analysis. Tfh cells were stained with FITC-labeled anti-CD4, PE-labeled anti-CXCR5, and APC anti-mouse CD279 (PD-1) antibodies (1:1000) (BD Biosciences) for detection. The resulting cell samples were examined with a FACSCalibur flow cytometer with cellQuest software (BD Biosciences).

### 4.10. Statistical Analysis

The data were expressed as the means ± standard error. The data were also subjected to analysis of variance and Tukey’s multiple comparison tests. *p*-values less than 0.05 were considered as statistically significant.

## 5. Conclusions

This study demonstrated that the oral feeding of P-CPP to mice attenuated the Th2-type induced allergic response in OVA-sensitized mice. Our findings reveal that the phosphate groups of CPP play an important role in the induction of Tre and Tfh cells.Thus, these results suggest that CPP phosphorylation would be a promising, safe and effective strategy in preventing IgE-mediated allergic diseases.

## Figures and Tables

**Figure 1 molecules-24-00738-f001:**
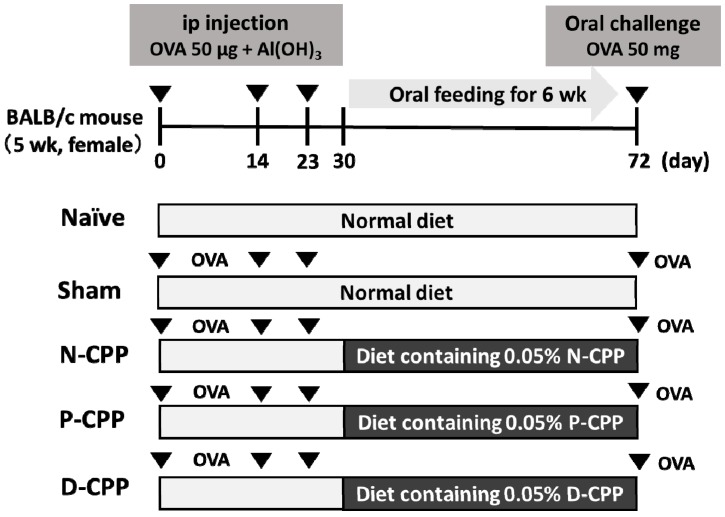
Scheme of in vivo experiment and systemic anaphylactic reactions post-oral challenge with ovalbumin (OVA). Female BALB/c mice were divided into five groups (*n* = 6). Naïve group was not sensitized, not orally challenged, and fed a commercial mice diet; sham-treated group was sensitized with OVA, orally challenged with OVA, not treated, and fed a commercial diet; native CPP-III (N-CPP), phosphorylated CPP-III (P-CPP), and dephosphorylated CPP-III (D-CPP) were sensitized and orally challenged with OVA and then fed a homemade diet containing 0.05% of N-CPP, P-CPP, and D-CPP, respectively. BALB/c mice were sensitized with 50 μg of OVA plus aluminum hydroxide Al(OH)_3_ as an adjuvant. Sensitization was repeated 14 days after the initial sensitization. After 6 weeks, oral challenge with 50 mg/mL OVA in PBS was conducted.

**Figure 2 molecules-24-00738-f002:**
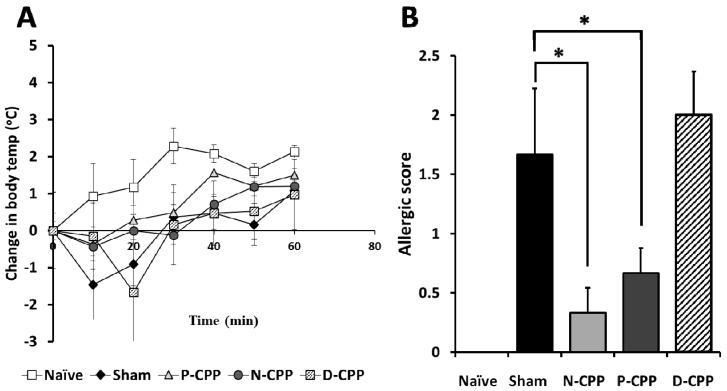
Effects of oral feeding with differently phosphorylated CPP-III on body temperature and allergic score following oral challenge with OVA in OVA-sensitized mice. OVA-sensitized mice were fed differently phosphorylated CPP-III for a period of 6 weeks, and the body temperature (**A**) and allergic score (**B**) were determined in the oral challenge test. Body temperature changes were recorded at 0, 10, 20, 30, 40, 50, and 60 min, while allergic scores and anaphylactic symptoms were evaluated and scored from 40–90 min after oral challenge. Data represent the mean ± standard error (SE) of individual mice in the group. * *p* < 0.05.

**Figure 3 molecules-24-00738-f003:**
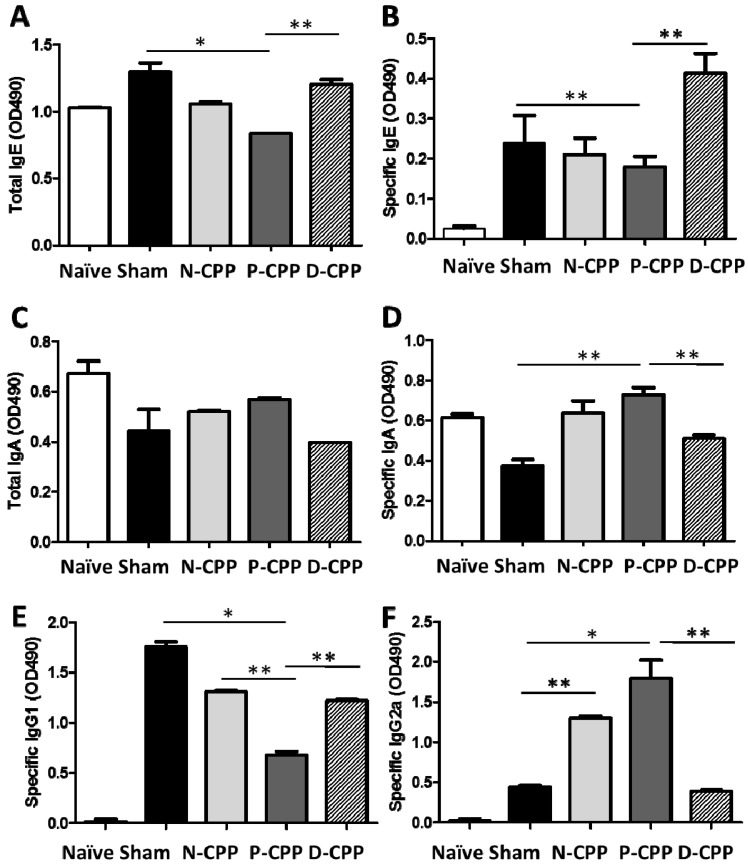
Effects of oral feeding with differently phosphorylated CPP-III on the immune response of OVA-sensitized mice. OVA-sensitized mice were fed differently phosphorylated CPP-III for a period of 6 weeks, and the serum levels of total IgE (**A**), OVA-specific IgE (**B**), total IgA (**C**), OVA-specific IgA (**D**), OVA-specific IgG1 (**E**), and OVA-specific IgG2a (**F**) were determined by ELISA. Data represent the mean ± SE of individual mice in the group. * *p* < 0.05; ** *p* < 0.01.

**Figure 4 molecules-24-00738-f004:**
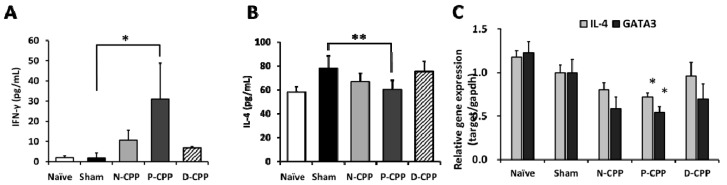
Effects of oral feeding with differently phosphorylated CPP-III on cytokine production and gene expression of OVA-sensitized mice. OVA-sensitized mice were fed differently phosphorylated CPP-III for a period of 6 weeks, and the collected spleen (SP) cells were incubated with PBS (unstimulated) for normal group and stimulated with 50 μg/mL (final concentration) of OVA for the control, N-CPP, P-CPP, and D-CPP groups. The levels of IFN-γ (**A**) and IL-4 (**B**) in the culture supernatant after incubation for 72 h were determined by sandwich ELISA. Relative mRNA expression levels of IL-4 and GATA-3 (**C**) were determined by qPCR (**C**). Data represent the mean ± SE of individual mice in the group. * *p* < 0.05; ** *p* < 0.01 vs sham group.

**Figure 5 molecules-24-00738-f005:**
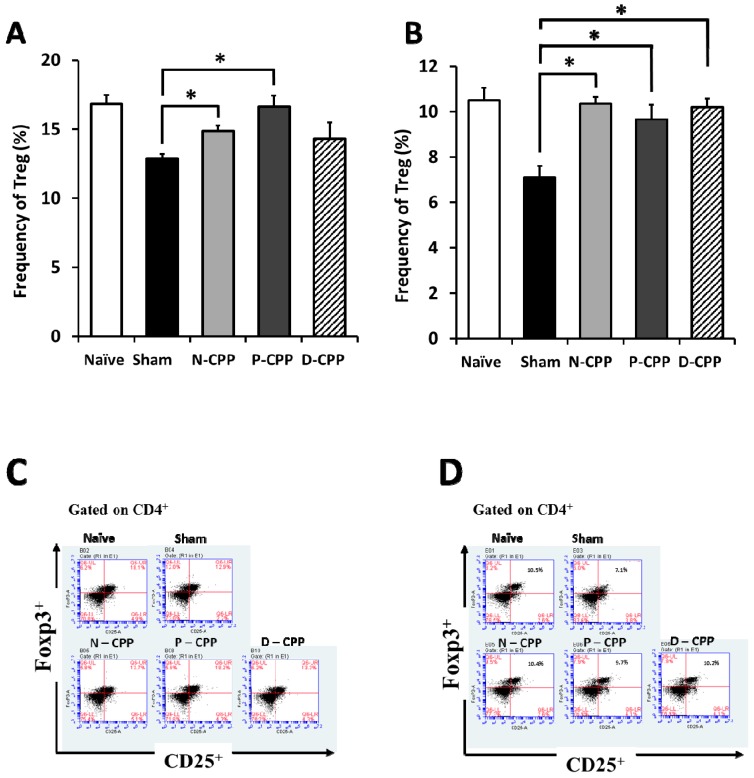
Effects of oral feeding with differently phosphorylated CPP-III on Treg cell population. Percentages of Treg cell populations in cultured SP (**A**) and payer’s patch (PP) (**B**). Representative FACS staining for SP (**C**) and from PP (**D**). Cells harvested from OVA-sensitized mice were determined by flow cytometry without stimulation with OVA. Data represent the mean ± SE of individual mice in the group. * *p* < 0.05 and dots represent cell proliferation.

**Figure 6 molecules-24-00738-f006:**
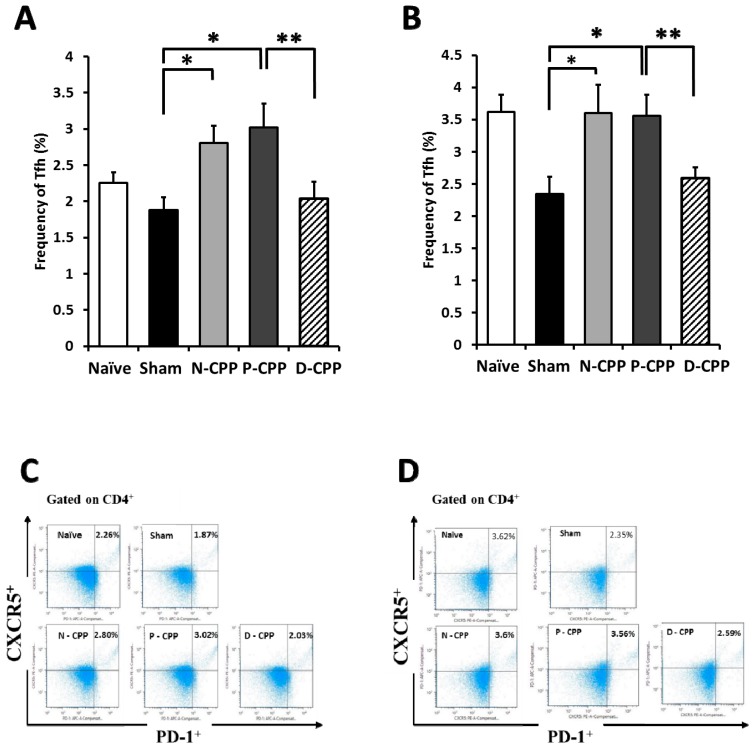
Effects of oral feeding with differently phosphorylated CPP-III on T follicular helper (Tfh) cell population. Percentage of Tfh cell populations in cultured SP (**A**) and PP (**B**). Representative FACS staining for SP (**C**) and PP (**D**). Cells harvested from OVA-sensitized mice were determined by flow cytometry without stimulation with OVA. Data represent the mean ± SE of individual mice in the group. * *p* < 0.05; ** *p* < 0.01 and dots represent cell proliferation.

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
