# Peer review of "Enhanced Anti-Allergic Activity of Milk Casein Phosphopeptide by Additional Phosphorylation in Ovalbumin-Sensitized Mice"

_molecules, 2019, doi:10.3390/molecules24040738_

Round 1

Reviewer 1 Report

The authors addressed all the comments adequately.

Author Response

We’d like to thank you for your time and effort devoted to reviewing our manuscript, and for the valuable comments.

Reviewer 2 Report

In this study, the authors investigated the anti-allergic activities of phosphorylation of casein phosphopeptide (CPP-III) in type-I allergic reactions. In an ovalbumin (OVA)-induced IgE-mediated allergy mouse model, they compared the anti-allergic activities of native CPP-III (N-CPP), additionally phosphorylated CPP-III (P-CPP) and dephosphorylated CPP-III (D-CPP). They found that oral administration of P-CPP suppressed anaphylactic symptoms, total and OVA specific IgE in mice. P-CPP skewed the balance from Th2 towards Th1 dominance, evidenced by lower levels of OVA-specific IgG1 and IL-4, and increased OVA-specific IgG2a and IFN-γ in P-CPP treated group. They also assessed the effects of P-CPP on regulatory T (Treg) cell and T follicular helper (Tfh) cell differentiation. They concluded that P-CPP treatment could attenuate allergen-specific IgE-modulated allergic reactions by inducing Tfh cells.

Still there are some questions that remain to be addressed:

1.     Please include representative FACS staining for Tfh cells and Tregs.

2.    In this study, the authors found that Th1 and Tfh cells were increased while Th2 cells were decreased. More tests of T helper cell-related cytokines are needed, such as TNF-α and IL-6 for Th1; IL-5 and IL-13 for Th2 and IL-21 for Tfh cells.

3.    To fully characterize the changes of Th2 and Tfh cells, the authors need to test their specific gene expression profile in sorted CD4+ T cells, such as cytokine IL-4 and IL-21, transcriptional factors GATA3, STAT6, Bcl-6, ICOS et al.

4.    In addition, please add the citation to the statement “Previous studies showed that Tfh can suppress Th2 type immune response with minimal negative effects on the Th1 response” in the discussion.

Author Response

We are very appreciative for the reviews provided by the reviewers of this manuscript. Their comments are positive and are of great help in revising this manuscript.  According to reviewer’s comments, we carefully revised the manuscript. Attached is the file of revised manuscript.

Reviewer 2

In this study, the authors investigated the anti-allergic activities of phosphorylation of casein phosphopeptide (CPP-III) in type-I allergic reactions. In an ovalbumin (OVA)-induced IgE-mediated allergy mouse model, they compared the anti-allergic activities of native CPP-III (N-CPP), additionally phosphorylated CPP-III (P-CPP) and dephosphorylated CPP-III (D-CPP). They found that oral administration of P-CPP suppressed anaphylactic symptoms, total and OVA specific IgE in mice. P-CPP skewed the balance from Th2 towards Th1 dominance, evidenced by lower levels of OVA-specific IgG1 and IL-4, and increased OVA-specific IgG2a and IFN-γ in P-CPP treated group. They also assessed the effects of P-CPP on regulatory T (Treg) cell and T follicular helper (Tfh) cell differentiation. They concluded that P-CPP treatment could attenuate allergen-specific IgE-modulated allergic reactions by inducing Tfh cells. Still there are some questions to be addressed

1. Please include representative FACS staining for Tfh cells and Tregs

Reply: Thank you for pointing this out. We added had the representative FACS staining images for Treg and Tfh cells in the Figure 5 and 6, respectively.

2. In this study, the authors found that Th1 and Tfh cells were increased while Th2 cells were decreased. More tests of T helper cell-related cytokines are needed, such as TNF-α and IL-6 for Th1; IL-5 and IL-13 for Th2 and IL-21 for Tfh cells.

Reply: Thank you for the suggestions. As you point out, their cytokine profiles will be more powerful data, but we provided the representative data of Th1 and Th2 cytokine in this work. To strength our data, we added the data of transcriptional factor GATA-3 as answer of your comment #3. Please confirm our answer of comment #3.

3. To fully characterize the changes of Th2 and Tfh cells, the authors need to test their specific gene expression profile in sorted CD4+ T cells, such as cytokine IL-4 and IL-21, transcriptional factors GATA3, STAT6, Bcl-6, ICOS et al.

Reply: Thank you for your constructive suggestion. We added some data concerning the gene expression profile of transcription factor GATA-3 and IL-4 cytokine in spleen cells. The data were added in the revised manuscript as Fig. 4C. These results strongly supported our conclusion that phosphorylation of CPP enhanced the attenuation of Th2 immune response. We revised some sentences in the Result, Discussion, Materials and Methods sections as well as References.

Revised sentences in Result section:

<Page 6, line 155-162>

To further elucidate and characterize the differentiation of naïve T cells to either Th1 or Th2 cells after oral treatment with P-CPP, we tested for the gene expression levels of the transcription factor GATA-3 and IL-4 cytokine in isolated SP cells. There was a significant reduction in expression of both GATA-3 and IL-4 mRNA in P-CPP fed mice as compared to N-CPP and D-CPP (P < 0.05). IL-4 is reported to induce B-cell class switching to IgE and also decreases Th1 cells production, while GATA-3 is selectively expressed in Th2 but not in Th1 and it also important in expression of IL-4 in T cells by transactivation of IL-4 promoter, the results suggest their suppression promoted the shift from Th2 immune response to Th1 immune response.

Revised sentences in Discussion section:

<Page 10, line 270-283>

Helper T cells are generally induced by different cytokines and controlled by transcription factors. They are reported to be classified into Th1 and Th2 determined by signature of cytokines produced [27], these T cells are known to maintain a well-balanced relations in the modulation of cytokine secretion to keep homeostasis in the host [4]. In order to characterize Th2 response we measured the effect of P-CPP ingestion on gene expression levels of the transcription factor GATA-3 and IL-4 cytokine. Both GATA-3 and IL-4 mRNA levels in total spleen cells were significantly inhibited by oral treatment with P-CPP. GATA-3 is a key transcription factor that is selectively expressed at high levels in Th2 cells [28]. It is conventionally regarded as a transcription factor that drives and regulates the differentiation of Th2 from naïve CD4+ lymphocytes [27,29], and it is reported to significantly inhibits the expression of Th1 related cytokine IFN-γ while upregulating gene expression of Th2 related IL-4 and IL-5 [30] Th2 related cytokines are the mediators of allergic inflammation. Previously Finotto et.al found out that the blockade of GATA-3 expression in the lung by oligonucleotide lead to the suppression of airways inflammation [31].

Added references

4.          Katayama, S.; Mine, Y. Quillaja saponin can modulate ovalbumin-induced IgE allergic responses through regulation of Th1/Th2 balance in a murine model. J. Agric. Food Chem. 2006, 54, 3271–3276, doi:10.1021/jf060169h.

27.    Wan, Y.Y. GATA3: A master of many trades in immune regulation. Trends Immunol. 2014, 35, 233–242, doi:10.1016/j.it.2014.04.002.

28.        Zheng, W.P.; Flavell, R.A. The transcription factor GATA-3 is necessary and sufficient for Th2 cytokine gene expression in CD4 T cells. Cell 1997, doi:10.1016/S0092-8674(00)80240-8.

29.        O’Garra, A.; Gabryšová, L. Transcription Factors Directing Th2 Differentiation: Gata-3 Plays a Dominant Role. J. Immunol. 2016, 196, 4423–4425, doi:10.4049/jimmunol.1600646.

30.        Ferber, I.A.; Lee, H.J.; Zonin, F.; Heath, V.; Mui, A.; Arai, N.; O’Garra, A. GATA-3 significantly downregulates IFN-γ production from developing Th1 cells in addition to inducing IL-4 and IL-5 levels. Clin. Immunol. 1999, doi:10.1006/clim.1999.4718.

31.        Finotto, S.; De Sanctis, G.T.; Lehr, H.A.; Herz, U.; Buerke, M.; Schipp, M.; Bartsch, B.; Atreya, R.; Schmitt, E.; Galle, P.R.; Renz, H.; Neurath, M.F. Treatment of allergic airway inflammation and hyperresponsiveness by antisense-induced local blockade of GATA-3 expression. J Exp Med 2001, 193, 1247–1260, doi:10.1084/jem.193.11.1247.

Revised sentences in Materials and Methods section:

<Page 13, line 419-431>

4.8. GATA-3 and IL-4 gene expression analysis by quantitative PCR (qPCR)

The total RNA was isolated using Invitrogen, TRizol reagent (Thermo Fisher Scientific, Inc., Waltham, MA, USA) in accordance with the manufacturer`s instructions. cDNA was synthesized from total RNA samples using ReverTra Ace (Toyobo, Osaka, Japan). Gene expression levels were quantified with a Kapa SYBR Fast qPCR kit (Kapa Biosystems, Woburn, MA, USA) and qPCR was performed with a Thermal Cycler Dice Real time system (Takara, Shiga, Japan). Fold changes in the relative mRNA expression level for each gene were normalized to that of housekeeping gene Glyceraldehyde-3-phosphate dehydrogenase (GAPDH). The expression in experimental groups was expressed as a fold of control using the comparative Ct method. Forward and reverse primer sequences were: GATA-3: 5`CTACCGGGTTCGGATGTAAGTC-3` (forward) and 5`GTTCACACACTCCCTGCCTTCT-3` (reverse); IL-4: 5`CGAAGAACACCACAGAGAGTGAGCT-3`(forward) and 5`GACTCATTCATGGTGCAGCTTATCG-3`(reverse); and GAPDH: 5`ACAACTTTGGCATTGTGGAA-3`(forward) and 5`GATGCAGGGATGATGTTCTG-3` (reverse).

4. In addition, please add the citation to the statement “Previous studies showed that Tfh can suppress Th2 type immune response with minimal negative effects on the Th1 response” in the discussion

Reply: Thank you for your suggestion. We added the following references in the discussion.

Revised sentences:

<Page 10, line 294-298>

Previous studies have also demonstrated that Tfh secreted IL-21 can suppress T-cell production of Th2 cytokines, mast cell degranulation and inhibit production of allergen specific IgE [33,34]. The inhibition of class switching to IgE by IL-21 was proven by production of high levels of IgE by IL-21R-deficient mice as compared to those with IL-21 receptor [35].

Added references

33.     Tamagawa-Mineoka, R.; Kishida, T.; Mazda, O.; Katoh, N. IL-21 reduces immediate hypersensitivity reactions in mouse skin by suppressing mast cell degranulation and IgE production. J. Dermatol. Sci. 2010, doi:10.1016/j.jdermsci.2010.10.001.

34.     Kemeny, D. The role of the T follicular helper cells in allergic disease. Cell. Mol. Immunol. 2012, 9, 386–389, doi:10.1038/cmi.2012.31.

35.     Hiromura, Y.; Kishida, T.; Nakano, H.; Hama, T.; Imanishi, J.; Hisa, Y.; Mazda, O. IL-21 Administration into the Nostril Alleviates Murine Allergic Rhinitis. J. Immunol. 2007, doi:10.4049/jimmunol.179.10.7157.

Reviewer 3 Report

Lebetwa et al. have evaluated the therapeutic effects of phosphorylated casein phosphopeptide (P-CPP) on egg allergy and they have shown that P-CPP is able to induce regulatory and follicular T cells. 

However, they have not evaluated the potential allergenicity or antigenicity of P-CPP before its evaluation as an immunomodulatory compound. Milk allergy is one of the most important food allergies and caseins (as well as their derived peptide fragments) have been reported to present a high sensitizing capacity. I think that author should assess the in vivo sensitizing capacity of P-CPP before evaluating its therapeutic effects on egg allergy. 

Follicular T cells generated on the spleen are going to rapidly migrate to the lymph node where they exert their immunological functions. The conclusions derived from data obtained in splenocytes should not be considered as representative of a real increase of this sort of cells. The potential increase of follicular T cells induced by P-CPP should be evaluated in the mesenteric lymph nodes. 

Author Response

We are very appreciative for the reviews provided by the reviewers of this manuscript. Their comments are positive and are of great help in revising this manuscript.  According to reviewer’s comments, we carefully revised the manuscript. Attached is the file of revised manuscript.

Reviewer 3

Lebetwa et al. have evaluated the therapeutic effects of phosphorylated casein phosphopeptide (CPP) on egg allergy and they have shown that P-CPP is able to induce regulatory and follicular T cells.

1. However, they have not evaluated the potential allergenicity or antigenicity of P-CPP before its evaluation as an immunomodulatory compound. Milk allergy is one of the most important food allergies and caseins (as well as their derived peptide fragments) have been reported to present a high sensitizing capacity. I think that author should assess the in vivo sensitizing capacity of P-CPP before evaluating its therapeutic effects on egg allergy. 

Reply: Thank you for your valuable comments. As you mentioned, cow’s milk allergy is one of the most important food allergies and the major protein is casein, which accounts for 80% of milk proteins and have a high sensitizing capacity. In this study, we observed no allergic symptoms in P-CPP fed mice, but our next investigation will need to focus on the in vivo sensitizing capacity of P-CPP. In this work, we added some sentences concerning the possibility of the sensitizing capacity of P-CPP as below.

Revised sentences:

<Page 11, line 307-315>

  Even though proteins and peptides exhibit these bioactive potentials, there is a possibility of exerting some allergic reactions. Information about their allergenicity is crucial in their all-inclusive evaluation. Reddi et al. demonstrated that CPP suppressed the release of histamine and tryptase from mouse mast cells, suggesting the attenuation of mast cell degranulation by CPP [42]. On the other hand, Bernard et al. when evaluating the influence of phosphorylation, on immunoreactivity of caseins found out that major phosphorylation sites in caseins are an important allergenic epitopes [43]. In this study, we observed no allergic symptoms in P-CPP fed mice, but further investigation will need to clarify the in vivo sensitizing capacity of P-CPP.

Added references

42.   Reddi, S.; Kapila, R.; Dang, A.K.; Kapila, S. Evaluation of allergenic response of milk bioactive peptides using mouse mast cell. Milchwissenschaft-Milk Sci. Int. 2012, doi:10.1055/s-0029-1200104.

43.     Bernard, H.; Meisel, H.; Creminon, C.; Wal, J.M. Post-translational phosphorylation affects the IgE binding capacity of caseins. FEBS Lett. 2000, doi:10.1016/S0014-5793(00)01164-9.

2. Follicular T cells generated on the spleen are going to rapidly migrate to the lymph node where they exert their immunological functions. The conclusions derived from data obtained in splenocytes should not be considered as representative of a real increase of this sort of cells. The potential increase of follicular T cells induced by P-CPP should be evaluated in the mesenteric lymph nodes

Reply: As you pointed out, the population of follicular T cells in the mesenteric lymph nodes would provide more reliable data to confirm the effects of P-CPP intake. In this study, we focused on the populations of the follicular T cells in spleen and Peyer’s patch because Peyer’s patch comprises the most important IgA induction site in the gut-associated lymphoid tissue. We will conduct further experiments concerning the population of follicular T cells in the mesenteric lymph nodes for our next paper according to your kind suggestion.

Round 2

Reviewer 2 Report

None

Reviewer 3 Report

No comments

This manuscript is a resubmission of an earlier submission. The following is a list of the peer review reports and author responses from that submission.

Round 1

Reviewer 1 Report

In this manuscript, Lebetwa and colleagues evaluated the possible suppressive effect of phosphorylated casein polypeptide -III against allegic immune responses in OVA-sensitized mice. 

Their findings support that treatment of OVA-sensitized mice with phosphorylated-CPP inhibited allergen-specific IgE-modulated allergic reactions via the induction of Treg and follicular Thelper cells, due to the presence of phosphate groups. 

Their data are well-sound, supported by appropriate methodology, and the manuscript well-written. 

However, there are certain points that need to be addressed:

How many times have the experiments (treatment of mice) been repeated?

It would be helpful if the authors provide a figure showing the gating strategy applied in the FACS experiments, maybe as supplementary material.

The manuscript need proof-reading for textual errors. 

Reviewer 2 Report

In this study, the authors investigated the anti-allergic activities of phosphorylation of casein phosphopeptide (CPP-III) in type-I allergic reactions. In an ovalbumin (OVA)-induced IgE-mediated allergy mouse model, they compared the anti-allergic activities of native CPP-III (N-CPP), additionally phosphorylated CPP-III (P-CPP) and dephosphorylated CPP-III (D-CPP). They found that oral administration of P-CPP suppressed anaphylactic symptoms, total and OVA specific IgE in mice. P-CPP skewed the balance from Th2 towards Th1 dominance, evidenced by lower levels of OVA-specific IgG1 and IL-4, and increased OVA-specific IgG2a and IFN-γ in P-CPP treated group. They also assessed the effects of P-CPP on regulatory T (Treg) cell and T follicular helper (Tfh) cell differentiation. They concluded that P-CPP treatment inhibits allergen-specific IgE-modulated allergic reactions by inducing Treg and Tfh cells and due to the presence of phosphate groups.

However, the validity and depth of this work are not strong enough to support their conclusions. Here are some comments:

1.     The function of additional phosphate group in the anti-allergic activity of CPP-III is not evident. In most of their results, including body temperature, allergic scores, specific IgE, IgA, IFN-γ, IL-4, Treg and Tfh cells, no difference was observed between P-CPP and N-CPP group. Therefore, it is not valid to conclude that P-CPP attenuates allergic response owing to high phosphorylation in CPP-III.

2.     The data in this study did not support the conclusion that P-CPP could attenuate type-I allergic response in OVA model mice through the induction Treg cells. In Figure 5, the frequency of Treg in N-CPP and D-CPP group was also higher than control group, and Treg in the N-CPP and D-CPP groups was even higher than P-CPP treated group. Thus, P-CPP attenuating allergic reaction by inducing Treg needs further investigation.

3.     In this study, the authors found P-CPP treatment induced Tfh cell differentiation. They assumed that amplified Tfh populations suppressed the allergic response by downregulating Th2 immune response or promoting IgA production. However, the authors need to provide more evidence to understand the role of Tfh cells in allergic reactions.

A number of recent reports have indicated a positive correlation between Tfh cells and allergy disease activity (Yao, Wang et al. 2018). Tfh cells play a critical role in promoting the production of IgE by secreting IL-4 (Kobayashi, Iijima et al. 2017, Kubo 2017, Dolence, Kobayashi et al. 2018).

4.    In addition, the description of some experiments is not clear. Such as in the experiment of evaluating the effects of P-CPP on Treg and Tfh cell differentiation, it is not clear whether SP and PP cells were stimulated with OVA in vitro before they tested.

Reviewer 3 Report

Well written manuscript with only minor changed required.

Line 118: total

Line 121-122: The sentence (Mice in the P-CPP treated group exhibited increase in levels of total IgA compared to the sham-treated, N-CPP, and D-CPP treated groups) and the first part of the next sentence in lines 122-123 (In contrast, specific IgA levels were significantly increased in P-CPP fed mice compared to sham-treated group), pretty much says the same thing, so remove one of them.

Reviewer 4 Report

In this work, Lebetwa and co-authors have showed that the treatment with phosphorylated casein phosphopeptide (P-CPP) is able to inhibit IgE-mediated allergic reactions in mice sensitized to ovalbumin by inducing regulatory and follicular T cells. Other biological activities have been previously described for CPP, although their anti-allergenic effects have been shortly studied.

However, since caseins have been described as major allergens of cow´s milk, the main limitation of this work is that the allergenicity of P-CPP must be studied and its immunomodulatory capacity has to be prior evaluated in the context of the cow´s milk allergy. Could authors show any result regarding the hypoallergenicity of P-CPP in vivo? Have they evaluated the effect of P-CPP in the treatment of milk allergy?

On the other hand, authors have chosen a systemic model of ovalbumin sensitization although they used an oral route of treatment and challenge. Have the effects of P-CPP been evaluated in an oral model of ovalbumin sensitization such as intragastric administration of ovalbumin together with cholera toxin?

Some conclusions of this article are based on the protective increased of serum level of ovalbumin-specific IgA and the splenic frequency of CXCR5+ CD4+ T cells. It has been previously described that specific IgA exerts a protective effect when it is released to the gut. Have authors evaluated the ovalbumin-specific IgA in feces? Regarding follicular T cells, due to their physiological migration, they should be studied in secondary lymphoid organs. Did authors also evaluate the increased presence of CXCR5+ PD1 CD4+ T cells in the mesenteric lymph nodes?

Minor comments

How many ovalbumin was used for intraperitoneal sensitization? In the line 82 authors indicated 50 µg although they wrote 100 µg in the line 299.

Figure 1 should indicate the number of mice used per group.

Since a score category has been used for clinical symptom description, authors should not indicate the allergic score per group as an average value (line 91) and the individual score per mouse should be used in figure 2.

In the line 309 authors indicated that PP cells were used for cytokine analyses, although figure 4 only showed that results regarding spleen cells. Please, could you explain it?

Since the percentages of cells in the PPs are no coherent with previously published results, were cells isolated from PP and spleen cells used for flow cytometry analysis after sacrifice or they were previously stimulated with ovalbumin in an ex vivo culture? Please, clarify this point.

The use of a gating strategy in figures 5 and 6 would help in the better understanding of these results.

Regarding the discussion, author should focus on the effect of hypoallergenic peptides in the treatment of food allergy. From the line 221 they discussed the effect of L. casei on food allergy. However a couple of reviews published in 2017 and 2018 have summarized a large number of studies that have studied immunomodulatory peptides for the prevention and treatment of food allergy. I think that they could be useful for this section of the discussion. On the other hand, from the line 243 authors discussed the importance of regulatory T cells in the resolution of food allergy. Could they hypothesize how P-CPP could induce antigen-specific regulatory T cells that help in the induction to ovalbumin tolerance?